# Arbuscular Mycorrhizae Alter Photosynthetic Responses to Drought in Seedlings of *Artemisia tridentata*

**DOI:** 10.3390/plants12162990

**Published:** 2023-08-19

**Authors:** Mathew Geisler, Sven Buerki, Marcelo D. Serpe

**Affiliations:** Department of Biological Sciences, Boise State University, 1910 University Drive, Boise, ID 83725, USA

**Keywords:** *Artemisia tridentata*, arbuscular mycorrhizae, drought, water potential, stomatal conductance, photosynthesis, water use efficiency

## Abstract

The establishment of *Artemisia tridentata*, a keystone species of the sagebrush steppe, is often limited by summer drought. Symbioses with arbuscular mycorrhizal fungi (AMF) can help plants to cope with drought. We investigated this possible effect on *A. tridentata* seedlings inoculated with native AMF and exposed to drought in greenhouse and field settings. In greenhouse experiments, AMF colonization increased intrinsic water use efficiency under water stress and delayed the decrease in photosynthesis caused by drought, or this decrease occurred at a lower soil water content. In the field, we evaluated the effect of AMF inoculation on colonization, leaf water potential, survival, and inflorescence development. Inoculation increased AMF colonization, and the seedlings experienced water stress, as evidenced by water potentials between −2 and −4 MPa and reduced stomatal conductance. However, survival remained high, and no differences in water potentials or survival occurred between treatments. Only the percentage of plants with inflorescence was higher in inoculated than non-inoculated seedlings. Overall, the greenhouse results support that AMF colonization enhances drought tolerance in *A. tridentata* seedlings. Yet, the significance of these results in increasing survival in nature remains to be tested under more severe drought than the plants experienced in our field experiment.

## 1. Introduction

Arbuscular mycorrhizal fungi (AMF) are obligate biotrophs that form associations with an estimated 72% of land plants [1]. These associations are widespread in terrestrial ecosystems, occurring in wetlands to deserts and the tropics to the low Arctic [2,3,4]. Numerous studies have shown that plants can increase their growth by establishing symbiotic associations with AMF [5]. The AMF provides mineral nutrients to the plant, particularly phosphorus (P) and other nutrients with low soil mobility, while plants provide carbohydrates and lipids to their fungal partners [5,6]. In addition, other beneficial effects of AMF on plants have been reported. These include less susceptibility to pathogens and increased tolerance to abiotic stresses caused by salinity, toxic metals, and drought [7,8,9,10].

For drought, in particular, results indicate that AMF can increase plant drought tolerance through various mechanisms [10,11,12]. Some of these are related to the role of AMF in nutrient uptake. As soil moisture declines, P uptake by AMF hyphae becomes more critical due to the more tortuous pathway of water movement to the root surface [13]. Also, growth promotions associated with higher nutrient acquisition can result in a larger root system, more apt to cope with drought [12]. Other mechanisms by which AMF help plants withstand drought, not necessarily related to mineral nutrition, are via reductions in oxidative stress and enhancements in root and soil hydraulic conductivity [14,15,16]. Water transport through the extraradical hyphae to the plant is often described as a means by which AMF alleviates drought stress. However, there are conflicting results about the significance of this pathway. Some studies indicate that water transport through extraradical hyphae is negligible compared to the direct uptake by the roots [17,18]. On the other hand, Li et al. [19] showed that colonization by *Rhizophagus intraradices* compensated for the absence of root hairs in a bald root barley mutant under mild water stress, thus suggesting that water uptake via AMF can affect plant water status. In addition, under drought, observations of higher water potential in mycorrhizal than non-mycorrhizal plants are not uncommon; this difference may be caused by the various mechanisms or effects described above [20,21,22].

A typical physiological response to water stress is decreased stomatal conductance (g_s_) [23]. A meta-analysis of more than 400 studies indicates that overall AMF increases g_s_ and that various factors affect the extent of this increase, including host species, AMF taxa, plant nutrient status, and experimental conditions [24]. Under well-watered conditions, mycorrhizal plants had, on average, 24% higher g_s_ than non-mycorrhizal ones, and the difference was greater under water stress [24]. Even though higher g_s_ increases water loss, the associated increase in CO_2_ assimilation might be advantageous [25]. The additional photosynthates resulting from higher g_s_ could support root growth in still-moist soil patches or sustain metabolism when more severe drought causes stomatal closure [26,27]. Gains in photosynthates can also result from an AMF effect on intrinsic water use efficiency (iWUE), the ratio of CO_2_ assimilation to g_s_ [23]. Enhancement in photosynthesis resulting from better mineral nutrition and reduction in oxidative damage in mycorrhizal plants tends to increase iWUE [14,28,29,30].

Independent of the mechanisms involved, there has been an increased interest in using AMF to mitigate plant water stress in crops and seedlings transplanted to forest plantations and disturbed natural habitats [31,32]. In Western North America, an ecosystem that has experienced widespread human disturbances is the sagebrush steppe [33]. In particular, the introduction of non-native annual grasses has increased wildfire frequency [34]. Fires tend to remove various native species, including a dominant and keystone shrub, *Artemisia tridentata* (big sagebrush, Asteraceae) [33,34]. Efforts to re-establish this species have been extensive and costly, with varied but generally low success rates [35,36]. Multiple factors have limited *A. tridentata* re-establishment, but a significant one is summer drought [36,37,38]. 

*Artemisia tridentata* forms associations with AMF, and results from pot and field experiments indicate that this symbiosis can improve its growth and seedling establishment [39,40,41]. However, neutral and adverse effects of AMF on the growth or establishment of *A. tridentata* seedlings have also been reported [35,42,43,44]. In addition, questions remain about the impact of AMF colonization on seedlings’ drought tolerance. Early work by Stahl et al. [45] showed that potted AMF seedlings withstood lower soil water potentials than non-mycorrhizal ones, indicating that AMF increased drought tolerance. However, to our knowledge, it is the only published study on this aspect of the AMF-*A. tridentata* symbiosis, and it is unknown how AMF affected g_s_, iWUE, and the plant water potential. Also, AMF effects on drought tolerance may differ between greenhouse and field-grown plants due to differences in soil volume and other abiotic and/or biotic factors [46,47]. 

To better understand the effect of AMF on *A. tridentata* seedlings’ responses to drought, we conducted three experiments, two in the greenhouse and one in the field, using native AMF as the inoculum. Although AMF have low host specificity and are typically mutualistic, their effects on, for example, nutrient uptake and plant growth vary depending on the particular AMF-plant taxa involved in the association [48,49]. In this regard, an important factor is whether the AMF that associate with the plant are native or exotic. Native AMF tend to be more beneficial than exotic ones [49]. This trend is thought to be caused by the reinforcement of mutualistic partnerships over a long coexistence and better fitness of native AMF to local edaphic and climatic conditions [49,50,51]. The greenhouse experiments aimed to analyze how changes in g_s_, photosynthesis, and iWUE induced by drought differ between non-mycorrhizal and mycorrhizal plants. In one experiment, we inoculated seedlings with isolated spores. In the other experiment, we used a less laborious method that would be more feasible for restoration purposes. It consisted of inoculating seedlings at the stage when they are typically outplanted with soil and roots from trap cultures. This method was also tested under field conditions to assess if it was adequate to increase AMF colonization over the levels resulting from AMF naturally occurring in the soil. In addition, we evaluated the effect of field inoculation on seedlings’ water status and survival. In the greenhouse, we hypothesized that AMF colonization would delay the decline in g_s_ and photosynthesis induced by drought and increase iWUE. These responses would indicate positive effects of AMF on coping with drought [10]. For the field experiments, we hypothesized that increasing AMF colonization would help maintain higher seedling water potentials during drought, resulting in higher survival. 

While this study mainly aimed to analyze AMF effects on seedlings’ responses to drought, other fungal endophytes can be present in *A. tridentata* roots. Of these, a group that appears to be common in the roots of *A. tridentata* and other native and exotic species in sagebrush habitats is the dark septate endophytes (DSEs) [52,53]. DSEs form a paraphyletic group within the Ascomycota, and their name reflects that their life cycle includes a stage with dark-melanized hyphae [54,55]. Although numerous studies have shown beneficial effects of DSEs and other septate root endophytes on plant growth and stress tolerance [56,57,58], neutral or negative impacts are also common [59]. This variety of possible effects could complicate the interpretation of responses to AMF-inoculated plants. For example, if septate fungi were present in lower abundance in non-inoculated than AMF-inoculated seedlings, a potential increase in drought tolerance in the latter could be caused by AMF, septate fungi, or both. Alternatively, the septate fungi could reduce the effects of AMF. To determine the extent to which differences in the abundance of septate fungi between non-inoculated and AMF-inoculated seedlings might explain hypothesized differences in plants’ responses to drought, we quantified their occurrence as an additional factor that could affect the results.

## 2. Results

### 2.1. Greenhouse Experiment 1: Inoculation with Spores

For this experiment, seeds of *Artemisia tridentata* ssp. *wyomingensis* (hereafter referred to as *A. tridentata*) were planted in 150 mL cone-tainers filled with a sterilized 3:2 sand-to-soil mix. The pots were thinned to one seedling per cone-tainer and randomly assigned to either the non-inoculated or inoculated treatment. The inoculated cone-tainers received an aqueous suspension containing spores extracted from trap cultures, while the non-inoculated cone-tainers did not receive spores. After inoculation, plants were grown in a greenhouse for eight months before being used to investigate AMF colonization and its effects on plant physiological responses to drought.

We measured colonization before and after the imposition of drought. Because the analysis of colonization involved harvesting the plants and using most of their root system, the seedlings measured before the drought were not the same as those measured afterward. Before withholding watering, the median value for total AMF colonization was 0 and 65.0% for non-inoculated and inoculated plants, respectively (*p* = 0.004) (Table 1). There was also a difference in arbuscular colonization, with a median of 0% in non-inoculated plants and 33.8% in inoculated ones (*p* = 0.015). Vesicles were less common, with a median of 9.0% in inoculated plants and 0 in non-inoculated ones (*p* = 0.002). We also measured colonization by septate endophytes. Total colonization by these fungi was low and not significantly different (*p* = 0.26) between non-inoculated and inoculated plants, with medians of 6.8 and 5.0%, respectively (Appendix A).

Following the drought period, inoculated plants maintained higher levels of AMF than non-inoculated ones (Table 1). Total AMF colonization post-drought in inoculated plants had a median of 48.2% and 0% in non-inoculated ones (*p* = 0.008). Differences were also detected for arbuscular colonization and the percentage of intersections with vesicles (Table 1). Albeit higher on average in inoculated seedlings, total and arbuscular colonization before the drought was not significantly different from that post-drought (*p* = 0.27 and 0.12, for total and arbuscular colonization, respectively). Also, withholding watering did not cause marked differences in DSE colonization (Appendix A).

To assess the changes in plant water status during the experiment, we measured leaf water potential (Ψ_l_) before withholding watering and after stomatal closure. In well-watered seedlings, there was no significant difference in Ψ_l_ between non-inoculated and inoculated plants, with mean values of −1.07 and −1.05 MPa, respectively (*p* = 0.84, Appendix A). After stomatal closure, Ψ_l_ was about 2.5 MPa lower than before withholding watering (Appendix A). The mean Ψ_l_ was −3.4 MPa for non-inoculated plants and −3.9 MPa for inoculated ones, but this difference was not significant (*p* = 0.4).

After withholding watering, the decline in CO_2_ assimilation (A), leaf transpiration (Tr), and stomatal conductance (g_s_) over time was not linear but followed approximately a negative sigmoidal curve (e.g., Appendix A). This response was modeled using Equation (1) (see Materials and Methods), which estimated the initial values of A, Tr, and g_s_, and the time since withholding watering when each parameter reached half its initial values (t_1/2_). The initial values of A, Tr, and g_s_ indicate the rates of these parameters when the plants were still well watered. There was no difference in these values between non-inoculated and inoculated plants (Table 2). As the drought progressed, values of A, Tr, and g_s_ markedly declined, which occurred sooner in non-inoculated plants. The average t_1/2_ for A in non-inoculated plants was 9.5 d and 13.7 d for inoculated ones (*p* = 0.037) (Figure 1). Similarly, the average t_1/2_ for Tr was 9.3 and 11.4 d for non-inoculated and inoculated plants, respectively (*p* = 0.034). The average t_1/2_ for g_s_ was 8.5 d in non-inoculated plants and 11.7 d for inoculated ones (*p* = 0.016) (Figure 1).

Like the gas exchange parameters, no difference in the photosystem II operating efficiency (ΦPSII) between non-inoculated and inoculated plants was detected under well-watered conditions (Table 1). After withholding watering, ΦPSII did not reach half its initial value during the drought, preventing the estimation of this parameter’s t_1/2_. 

As g_s_ decreased with water stress, the intrinsic water use efficiency (iWUE) tended to increase until g_s_ reached values of about 0.05 mol m^−2^ s^−1^. Below this value, iWUE was erratic or markedly decreased. Based on these results, we only analyzed the relationship between g_s_ and iWUE from the maximal g_s_ measured to 0.05 mol m^−2^ s^−1^. Within this range, there was a significant interaction between g_s_ and inoculation on iWUE (*p* = 0.0006). The increase in iWUE with decreasing g_s_ was somewhat more marked in inoculated than non-inoculated plants (*p* = 0.008) (Figure 2).

### 2.2. Greenhouse Experiment 2: Inoculation with Soil and Roots from Trap Cultures

In this experiment, we tested a different approach to inoculate the seedlings using the potting substrate and roots from trap cultures instead of spores only. Ten months old seedlings, like those typically used in outplantings [60], were transferred to 656 mL pots filled with a sterilized 3:2 sand–soil mix (non-inoculated seedlings) or the same mix supplemented with material from the trap cultures (inoculated seedlings). In addition, the non-inoculated seedlings were drenched with an AMF-free microbial wash [61]. After transplanting to the pots, plants were grown in a greenhouse for six months. At this time, the plants were used to conduct a completely randomized factorial combination experiment consisting of two watering treatments (well-watered and drought-stressed) and two inoculation treatments (non-inoculated and inoculated plants). The well-watered plants received water to pot capacity every other day. For the drought treatment, drought was imposed by withholding watering until stomatal conductance was minimal for about one week. 

Before withholding watering, the median value for total AMF colonization was 0 and 20.6% for non-inoculated and inoculated plants, respectively (*p* = 0.008, Table 3). The latter had medians for arbuscular colonization and vesicles of 6.9% and 2.0%; although low, these values were higher than those in non-inoculated plants (Table 3). After the drought, differences in colonization were similar or somewhat larger than before withholding watering, with median values for total AMF colonization of 1.55 and 32.3% for non-inoculated and inoculated plants, respectively (*p* = 0.004, Table 3). Albeit smaller, there were also differences in arbuscular colonization and the percentage of intercepts with vesicles, with medians of 0 in non-inoculated plants and 6.3 and 2.0% in inoculated ones. In plants kept well-watered, differences in colonization at the end of the experiment were similar to those exposed to drought (Table 3).

We also used a molecular sequencing approach to ascertain the AMF taxa colonizing the inoculated plants. Based on comparisons to sequences in datasets from fungal taxonomy [62,63], the taxa forming symbioses with *A. tridentata* roots were within the families Glomeraceae (163 amplicon sequence variants, ASVs) and Diversisporaceae (13 ASVs) (Figure 3). These ASVs have been deposited at the National Center for Biotechnology Information (NCBI) GenBank under accession numbers OR354512-OR354687. The Glomeraceae included taxa within the genera *Rhizophagus* and *Funneliformis* and taxa only identified to the family level (Figure 3). The taxa within the Diversisporaceae were in two genera, *Diversispora* and *Otospora*.

In this experiment, using soil and roots from the trap cultures for inoculation or applying an AMF-free microbial wash to the non-inoculated plants increased colonization by septate fungi. Colonization by these fungi was highly variable but higher overall than in experiment 1, particularly for samples harvested after drought imposition or at the experiment’s end when median values ranged from 19% to 34% (Appendix A). However, differences in colonization by septate fungi before (*p* = 0.06) or after the drought (*p* = 0.33) were not significant between non-AMF-inoculated and AMF-inoculated plants.

The effect of inoculation on A, Tr, g_s_, and ΦPSII was investigated in well-watered and drought-stressed plants. In plants kept well-watered, there were daily variations in A, Tr, g_s_, and ΦPSII within a plant, but without a trend of increases or decreases in these parameters over time (e.g., Appendix A). Given these results, we estimated the average of the A, Tr, g_s_, and ΦPSII measurements made for each plant during the 60 days of the experiment and used these averages to compare non-inoculated vs. inoculated plants. Appendix A summarizes the results of these comparisons; under well-watered conditions, inoculation did not affect any of the physiological parameters measured. 

As noted earlier, for plants exposed to drought, the initial values of A, Tr, g_s_, and ΦPSII estimated using Equation (1) indicate the rates of these parameters when the plants were still well-watered. We did not detect differences in these initial values between non-inoculated and inoculated plants (Table 4), further indicating that inoculation did not affect A, Tr, gs, or ΦPSII under well-watered conditions. As the rate of these parameters declined with drought, there was considerable variability between plants in the decline rates. However, this variability was not associated with the inoculation treatment. For a particular parameter, the t_1/2_ of non-inoculated plants was not significantly different from that of inoculated ones (Figure 4A).

Part of the variation in t_1/2_ was likely attributed to dissimilarities in total transpiration due to differences in leaf area or transpiration rates. To account for this variability, we also analyzed physiological responses as a function of the ratio between pot weight and weight at pot capacity (rPC). In particular, we used Equation (2) (see Materials and Methods and Appendix A) to estimate the ratio at which the physiological parameters measured declined to half their initial values (rPC_1/2_). Variation in rPC_1/2_ occurred within treatment (Figure 4B). Nevertheless, for the four parameters measured, rPC_1/2_ in non-inoculated plants was significantly higher than that of inoculated ones (*p* < 0.05). Thus, in non-inoculated plants, the decline in g_s_, CO_2_ assimilation, Tr, and ΦPSII occurred at a higher soil water content than in inoculated ones.

Due to the partially destructive nature of Ψ_l_ measurements, we did not follow changes in Ψ_l_ in individual plants. Instead, Ψ_l_ was determined in randomly selected plants throughout the experiment. Plotting these Ψ_l_ values against (1-rPC) shows that initially, decreases in rPC had little effect on Ψ_l_ and that plant water status was similar for non-inoculated and inoculated plants. However, as rPC continued declining, the reduction in Ψ_l_ tended to occur at higher rPC in non-inoculated than inoculated plants (Figure 5A), paralleling the results observed with gas exchange and chlorophyll fluorescence parameters. Before determining Ψ_l_ in the selected plants, we also measured their g_s_. As shown in Figure 5B, mycorrhization did not seem to affect the relationship between Ψ_l_ and g_s_.

For iWUE, the results were similar to those in experiment 1. iWUE increased with decreasing g_s_ (*p* < 0.00001), and there was a significant interaction between g_s_ and inoculation on iWUE (*p* = 0.03). The decrease in g_s_ increased iWUE more in inoculated than non-inoculated plants (*p* = 0.04) (Figure 6).

### 2.3. Field Experiment

This experiment was conducted in Kuna Butte, ID, USA (43° 26′ 47.32″ N, 116° 26′ 48.61″ W, 965 m), starting in October 2019. At this time, we transplanted 300 ten-month-old seedlings and randomly assigned them to one of two inoculation treatments, control and inoculated. In the control treatment, the seedlings were planted without inoculum. In contrast, 500 mL of soil and roots from the trap cultures were placed beneath and around each seedling for the inoculated treatment. We evaluated the effects of inoculation and summer drought on the following response variables: fungal colonization of roots, plant survival, leaf water potential, stomatal conductance, and percent of plants with flowers. 

Seedlings for colonization analysis were collected in June and October 2020, eight and twelve months after transplanting. For total AMF, arbuscules, and vesicles, two-way ANOVA indicated no significant interaction between inoculation treatment and sampling time on these variables (Appendix A). Sampling time only affected total AMF colonization (*p* = 0.02); total AMF colonization was higher in samples collected in the spring (35.8 ± 3.4%) than in the fall (21.1 ± 4.8). Adding trap culture inoculum beneath the seedlings increased total AMF and arbuscular colonization but did not affect the frequency of vesicles (Table 5). When averaged over the two sampling times, total AMF colonization was 16.9 and 39.9% for non-inoculated and inoculated seedlings, respectively (*p* = 0.0003). For arbuscular colonization, a significant difference was only apparent for the spring samples (Table 5). However, the combined results from spring and fall samples indicate that overall inoculation increased arbuscular colonization from 4.8% to 13.4% (*p* = 0.016). In addition to AMF, the roots were colonized by septate fungi. Total colonization by these fungi ranged from 16 to 54%. Neither the addition of trap culture inoculum nor sampling time or their interaction significantly affected septate fungi colonization or the presence of microsclerotia (Appendix A).

The observed differences in AMF colonization did not impact survival. Seedling mortality was negligible and independent of the inoculation treatment (Appendix A). In December 2020, approximately one year after transplanting, the survival for non-inoculated seedlings was 95.9%, and that for those inoculated was 97.2% (*p* = 0.5). Furthermore, by July 2020, many plants had inflorescences. The percentage of plants with inflorescences was 41.0 and 50.0% for the non-inoculated and inoculated seedlings, respectively, but the difference was not significant (*p* = 0.20). Survival remained high during the subsequent winter, spring, and summer (Appendix A). In late October 2021, the survival rate for non-inoculated and inoculated plants was 92.2 and 95.0%, respectively (*p* = 0.32). Also, in July 2021, many plants were in bloom, and the percentage of lived plants with inflorescences was lower for the non-inoculated ones (47.2%) than those inoculated (63.0%) (*p* = 0.02).

Although the plants had high survival rates, weather conditions suggested they had to cope with drought (Appendix A). During 2020, precipitation was minimal between early July and November. This lack of rainfall markedly decreased soil moisture in at least the upper 20 cm of the soil (Appendix A). Due to lower precipitation and higher temperatures than in 2020, the summer of 2021 also presented conditions conducive to drought stress (Appendix A). To characterize the degree of water stress the plants experienced, we measured midday Ψ_l_ during the summer and fall of 2020 and the midday Ψ_l_ and stomatal conductance during the spring and summer of 2021. 

From mid-summer to fall of 2020, average Ψ_l_ remained relatively constant between −2 and −3 MPa (Figure 7A). Most variation in Ψ_l_ was between plants, without differences between non-inoculated and inoculated seedlings. In 2021, we began the midday Ψ_l_ measurements in the spring, when water potentials were high with median values above −1 MPa (Figure 7A). Subsequently, Ψ_l_ gradually declined until late July and remained relatively constant between −2 and −2.5 MPa for the next month. Similar to 2020, inoculation did not affect Ψ_l_ since changes in this variable were similar in non-inoculated and inoculated plants. The seasonal decrease in Ψ_l_ was correlated with a reduction in g_s_ that, like Ψ_l_, was not affected by the inoculation treatment (Figure 7B).

## 3. Discussion

### 3.1. Greenhouse Experiments

The main objectives of the greenhouse experiments were to generate differences in AMF colonization between non-inoculated and inoculated seedlings and to characterize the effect of these differences on g_s_, photosynthesis, and iWUE. AMF colonization was minimal in non-inoculated seedlings, while it had median values of about 50 and 30% in seedlings inoculated with isolated spores and trap culture material, respectively. These differences in colonization did not affect leaf gas exchange, ΦPSII, and iWUE under well-watered conditions. In contrast, as drought developed, decreases in g_s_, CO_2_ assimilation, and ΦPSII occurred later or at a lower soil water content in inoculated than non-inoculated seedlings. Furthermore, as g_s_ decreased with water stress, iWUE increased more in inoculated plants. The effects of AMF colonization on prolonging photosynthesis and enhancing iWUE under drought agree with two of our stated hypotheses.

In addition to AMF, *A. tridentata* roots were colonized by septate fungi. In the experiment using isolated spores, colonization by these fungi was very low (Appendix A). Therefore, possible effects of septate fungi on the seedlings or AMF were likely minimal. For the experiment using soil and roots from trap cultures, the impact of septate fungi could have been more significant because colonization reached median values of about 20%. However, no significant differences in septate fungal colonization were detected between non-inoculated and AMF-inoculated seedlings (Appendix A). Consequently, differences between these treatments in iWUE and photosynthesis as the soil dried out cannot entirely be attributed to the presence of septate fungi.

While the two greenhouse experiments indicated that AMF affected the seedlings’ responses to drought, there were differences in the results. Significant effects on t_1/2_ were only observed in the experiment using isolated spores. Discrepancies between the two experiments may partly be attributed to differences in the level of AMF colonization (Table 1 and Table 3). Higher AMF colonization in the first experiment likely resulted in a larger effect of these fungi on photosynthesis and g_s_, thus allowing the detection of differences in t_1/2_ despite variations between plants. This idea is consistent with the results of Augé et al. [24]; their meta-analysis of stomatal responses to AMF revealed that the AMF effect on g_s_ was about ten times higher in heavily than lightly colonized plants. 

Apart from differences in colonization, the lack of an effect on t_1/2_ in the second experiment may reflect that time since withholding watering was not an accurate indicator of the drought stress the plants were experiencing. Due to differences in the density of short lateral shoots or their compacted-arranged leaves, the seedlings varied in leaf area despite their similarities in height. In addition, the plants showed dissimilarities in their transpiration rates independent of treatment (Table 4). Differences in leaf area and transpiration rates likely caused different water depletion rates and water potential decrease between the pots in a treatment, leading to considerable variability in g_s_ and other parameters when plotted against time. The ratio of percent pot weight to weight at pot capacity (rPC) is a more direct indicator of soil water depletion and water stress than time [64,65]. When considering this ratio as the independent variable, the gas exchange parameters and ΦPSII decline occurred at lower rPCs in mycorrhizal than non-mycorrhizal plants (Figure 4). Thus, in inoculated plants, the decrease in g_s_, CO_2_ assimilation, Tr, and ΦPSII happened at a lower soil water content than in non-inoculated ones. 

Data similar to ours showing transpiration and photosynthesis declining at a lower soil water content in mycorrhizal than non-mycorrhizal plants have been reported in other studies [17,66]. Various mechanisms can be responsible for such an effect. AMF can increase osmotic adjustment in plants exposed to drought [67,68]. Through opposite effects on plant Ψ and turgor, osmotic adjustment can prolong water uptake and photosynthesis as the soil dries out [69]. Although we did not measure the osmotic potential of the plants, a higher osmotic adjustment in inoculated than non-inoculated plants does not appear to explain our results. For example, in inoculated plants, the decrease in g_s_ occurred later or at lower rPc than in non-inoculated plants. If osmotic adjustment had caused these effects, the reduction in g_s_ would have happened at lower Ψ_l_ in mycorrhizal plants. However, this was not observed (Figure 5B). 

Another means by which AMF can affect plant water status is via changes in soil and plant hydraulic conductivity [16,70]. Bitterlick et al. [71] showed that mycorrhizal tomato plants reduced transpiration under increasing drought at lower potting substrate Ψ than non-mycorrhizal ones. Interestingly, the plant Ψ at which transpiration decreased was the same for AMF and non-AMF plants [71]. These results were attributed to the extraradical hyphae, which presumably enhanced liquid continuity through air gaps, thereby increasing the potting substrate’s hydraulic conductivity. The notion that AMF increase hydraulic conductivity seems consistent with our results. Compared to non-inoculated plants, AMF colonization decreased the rPC_1/2_ of transpiration and the rPC at which the leaf Ψ declined (Figure 4B and Figure 5A). Since rPC is related to soil Ψ, these results suggest that AMF plants maintained transpiration and leaf Ψ with a narrower gradient in the soil to leaf Ψ. Based on fundamental principles of water transport, such a situation would require an increase in soil or plant hydraulic conductivity [23,70]. 

While the effect of AMF on delaying t_1/2_ or lowering rPC_1/2_ is intriguing, several questions remain about its impact on coping with drought. Postponing stomatal closure with increased drought severity can be a disadvantage if it decreases plant Ψ to levels causing xylem embolism and, ultimately, death by hydraulic failure [72,73]. However, non-inoculated and inoculated plants showed a similar relationship between Ψ_l_ and g_s_ (Figure 5B), suggesting that prolonged photosynthesis under water stress did not increase the risk of xylem embolism in the latter. Another question is the extent the additional water extracted from the pots by mycorrhizal plants before stomatal closure can increase drought tolerance. This amount was relatively small, between 40 to 50 mL. This estimate comes from the weight at pot capacity (~1300 g) times the difference in rPC_1/2_ between non-mycorrhizal and mycorrhizal plants (~3 to 4%). Overall, 40 to 50 mL is a trivial amount to cause changes in the physiology of the plants under field conditions. However, water uptake can occur over a much-extended soil volume in the field. Consequently, a similar phenomenon of soil water depletion in natural settings is likely to result in the extraction of larger volumes of water. The extent of this effect needs to be determined. Still, if it helps the plants prolong photosynthesis without decreasing plant Ψ, it may help maintain water homeostasis. The plant could use the additional photosynthates to grow fine roots and hyphae toward moist soil patches, facilitating water uptake and preventing drought-induced damage and death. 

A consistent result in the two greenhouse experiments was that AMF did not affect the gas exchange parameters measured nor ΦPSII under well-watered conditions. Enhancements in photosynthesis by AMF are often associated with improved nutrient uptake, particularly P [5,13]. The presence of arbuscles suggests that the plants took P via mycorrhizae because arbuscules tend to collapse without P transport [74]. However, a remaining question is whether AMF colonization increased P uptake and other nutrients. In response to AMF colonization, some plants downregulate direct P uptake through the root epidermis and root hairs, resulting in mycorrhizal plants having a P concentration similar to non-mycorrhizal ones [75]. It is unknown the extent that *A. tridentata* behaves in this manner. Still, such behavior would have resulted in comparable mineral nutrient concentrations and, therefore, a lack of an effect on metabolic processes such as photosynthesis. Alternatively, mycorrhizal plants may have taken more P than non-mycorrhizal ones. However, in the latter, mobilization from older leaves could have maintained nutrient levels, resulting in a lack of differences in their concentrations in the relatively younger leaves where we conducted the measurements. 

Independent of the reasons involved, the lack of an effect of AMF on the rate of photosynthesis and g_s_ in well-watered plants explains the similarities in iWUE between non-inoculated and inoculated plants under these conditions. In contrast, as g_s_ declined with water stress, iWUE tended to be higher in inoculated plants. In a previous study, we observed a similar trend in *A. tridentata* seedlings growing outdoors [41]. Based on foliar ^13^C/^12^C isotope ratios, AMF colonization increased the iWUE of spring-outplanted seedlings that experienced summer drought but not on fall-outplanted seedlings sampled the following spring and much less exposed to drought [41]. In other species, symbioses with AMF are known to increase the formation of antioxidants, particularly in plants experiencing water deficits [76,77]. This increase dampens the buildup of reactive oxygen species that occurs under drought and diminishes the damaging effects of these molecules on photosynthesis and other processes [78,79]. Similar responses in mycorrhizal *A. tridentata* seedlings perhaps mediated their higher increase in iWUE with water stress. 

### 3.2. Field Experiment

The results from the field experiment showed that inoculation with sand–soil mix and roots from trap cultures increased colonization over the levels caused by AMF naturally occurring in the soil; the latter estimated from the extent of colonization of non-inoculated seedlings. The difference in AMF colonization between inoculated and non-inoculated seedlings was about 23% and 8% for total AMF and arbuscular colonization, respectively. These differences represent more than 100% increases over the colonization levels without inoculation. In *A. tridentata* and other species, similar increases in colonization have been associated with enhancements in growth, nutrient concentrations, tolerance to abiotic stresses, or survival [41,80,81]. Thus, the magnitude of the observed colonization increase can be of biological significance and is typically indicative of successful mycorrhizal inoculation [82,83]. 

Notwithstanding the observed difference in colonization and contrary to our hypothesis, we did not detect differences in Ψ_l_, even during the more severe drought period from mid-summer to early fall. Several studies have reported that AMF colonization contributes to maintaining higher water potentials during water deficits [21,22,67,84], which agrees with our greenhouse experiment (Figure 5A). However, these studies were in potted plants, comparing non-mycorrhizal vs. mycorrhizal plants and using watering regimens where drought developed more rapidly or lasted a shorter period than that experienced by *A. tridentata* seedlings in the field. The presence of AMF in the non-inoculated treatment may have reduced differences between treatments. In addition, the prolonged water stress and higher summer temperatures in the field (Appendix A) could have triggered different responses than those in potted plants in the greenhouse. One such plausible difference is in the extent of leaf abscission; *A. tridentata* is semi-deciduous, dropping many leaves as drought and heat intensify during the summer [85]. Although we did not quantify leaf drop, the extent of its occurrence seemed higher in the field than in the greenhouse experiments. Leaf drop combined with a gradual decrease in stomatal conductance of the persistent leaves may have made field plants less dependent on AMF for water balance.

Like Ψ_l_, inoculation did not affect survival; one year after outplanting, survival was above 95% for both treatments. This high survival rate was likely related to the ability of most plants to maintain Ψ_l_ above −3 MPa. While low, the plants’ water potentials during the summer were above those that cause death by hydraulic failure in this species, which typically occurs at water potentials between −4 and −8 MPa [86]. Also, the attained survival was higher than that reported in other outplantings [35,36,41]. Weather conditions cannot entirely explain the high rates of survival. The weather data indicates that precipitation and temperature during the spring and summer of 2020 (Appendix A) were typical for the region [87]. In addition, except for the planted seedlings and a few rabbitbrush plants, all the other vegetation dried out in early to mid-summer, indicating severe water scarcity. 

Factors that could have contributed to the successful establishment are deep watering, using metal-mesh protectors, and septate fungi. Water was only added immediately after outplanting and two weeks later through a PVC tube inserted in the soil. This method perhaps facilitated the growth of deeper roots during winter, giving access to moister soil layers in summer. Also, in this experiment, each seedling was within a metal-mesh protector. In a previous study [88], we observed that these protectors minimized herbivory and increased survival compared to Vexar protectors typically used in many outplantings. High survival rates could also reflect biotic characteristics of the site. Various reports indicate that septate fungi can increase tolerance to biotic and abiotic stresses [57]. Given the abundance of septate fungi in *A. tridentata* roots, such effects would have improved survival. 

In addition to the high survival rate, a somewhat unexpected result of the study was the development of inflorescences in many of the transplanted seedlings. Moreover, during the second year, the percentage of plants with inflorescences was higher in the inoculated than non-inoculated treatment, potentially impacting fecundity. The higher levels of AMF colonization in the inoculated seedlings may have caused somewhat better nutrient content or iWUE in the former, promoting flowering [89,90,91]. Also, the symbiosis can alter metabolic pathways throughout the plant affecting secondary metabolite production and hormonal balance [92,93,94]. Some of these changes have been linked to improved reproductive fitness and, in our study, could have also been responsible for the increase in the number of plants with inflorescences [94,95,96]. The ecological consequence of this last effect is unclear because as the plants age, most, if not all, will flower. Nevertheless, seeds from plants that bloom at a younger age might have a better chance of seedling recruitment in sites recovering from fires due to lower intraspecific competition or higher availability of favorable microsites [97,98].

## 4. Materials and Methods

### 4.1. Greenhouse Experiment 1: Inoculation with Extracted Spores

#### 4.1.1. Fungal and Plant Material

To produce mycorrhizal inoculum, we collected silty-loam soil from a sagebrush community near Kuna Butte, Idaho (43°26.161′ N, 116°25.848′ W, 908 m a.s.l.). This soil was mixed in a 2:3 ratio with sterilized sand. Subsequently, the AMF in this mix were multiplied in trap cultures using *Plantago lanceolata* as the host following the procedure described in the International Collection of Vesicular Arbuscular Mycorrhizal Fungi website [99]. After three cycles of trap culture cultivation, spores were extracted from these cultures by wet sieving and sucrose gradient centrifugation [100]. Then, the spores were surface-sterilized in 0.5% sodium hypochlorite for ten minutes, rinsed in sterile water, and stored at 4 °C in an aqueous solution containing 200 mg L^−1^ streptomycin and 100 mg L^−1^ gentamycin. Spores remained in this solution until the day of inoculation, which occurred within a month. On this day, the spores were resuspended in de-ionized water. Previous studies indicated that the trap cultures contained a mixture of AMF within the Glomeraceae family [101]. The Wyoming big sagebrush seeds used in this study (*Artemisia tridentata* ssp. *wyomingensis*, referred to as *A. tridentata*) were supplied by the Bureau of Land Management, and they had been collected from several mother plants throughout the Morley Nelson Snake River Birds of Prey National Conservation Area in southwestern Idaho, USA. This conservation area is about 200 hectares, and its nearest point is within 3 km of the site where we collected the soil for the trap cultures. 

#### 4.1.2. Growing Conditions, Experimental Approach, and Data Collection

This experiment was conducted at the Boise State University Research greenhouse (Boise, ID, USA), where *Artemisia tridentata* seeds were planted in about 30 150 mL cone-tainers (SC10R-Ray Leach, Stuewe & Sons, Inc., Tangent, OR, USA) filled with a 3:2 sand to soil mix, which had been autoclaved twice for 1 h. The soil used in this experiment was collected at the same sagebrush steppe community in Kuna Butte, Idaho (43°26.161′ N, 116°25.848′ W), where we collected the soil to start the trap cultures. The soil at this site is Power-McCain silty loam, classified as fine-silty, mixed, superactive, mesic Xeric Calciargids [102]. Before mixing it with sand, the soil was screened through a 1 mm mesh to remove leaf litter and roots. Two months after seeding, the pots were thinned to one seedling per cone-tainer and randomly assigned to either the non-inoculated or inoculated treatment. After extracting and resuspending spores from trap cultures, we estimated their density by counting spores in aliquots of the suspension. Inoculated cone-tainers received a volume of aqueous suspension to provide about 700 spores. The latter were placed 6 to 7 cm from the soil surface, while the non-inoculated cone-tainers did not receive spores. While adding dead spores to the non-inoculated treatment would have been a more precise control, we did not pursue this approach due to the laborious nature of spore production and collection. Also, preliminary work suggested that the small amount of organic material resulting from adding dead spores would not affect the seedlings’ responses to drought. After inoculation, plants were grown in a greenhouse for eight months. Since seeding and throughout the experiment, the plants were under a 15 h photoperiod with day/night conditions of 23/18 ± 3 °C. Until the beginning of the drought treatment, the sand/soil mix was kept close to field capacity and fertilized monthly with a 1/8 strength Hoagland’s solution. 

Eight months after inoculation, we used five non-inoculated and five inoculated *A. tridentata* seedlings to analyze the effect of AMF colonization on plant physiological responses to drought. These seedlings were similar in size, about 10 cm in shoot height. Drought was imposed by withholding watering, and subsequently, we made gas exchange measurements of the plants every day until stomatal conductance was minimal (<than 0.03 mol m^−2^ s^−2^, which was when net photosynthesis started to become negative) for three consecutive days. The parameters measured daily were CO_2_ assimilation per unit leaf area (A), transpiration per unit leaf area (Tr), stomatal conductance (g_s_), and photosystem II operating efficiency (ΦPSII). These parameters were measured using a LI-6400-40 leaf chamber fluorometer connected to a LI-COR LI-6400XT portable photosynthesis system (LI-COR Inc., Lincoln, NE, USA). The leaves were arranged to cover the leaf chamber area fully. Net photosynthesis, Tr, and g_s_ were measured at an incoming airflow of 200 µmol s^−1^, a CO_2_ concentration of 400 µmol mol^−1^, ambient temperature, and 500 µmol m^−2^ s^−1^ light intensity. Values of A and Tr were recorded after the CO_2_ assimilation rates and stomatal conductance values became stable, and the infrared gas analyzer was matched before each measurement. After the gas exchange measurements were completed, ΦPSII was determined in the same leaves by measuring the steady-state fluorescence (F’) and the maximal fluorescence (Fm’). The latter was measured following a light-saturating pulse of 8000 µmol m^−2^ s^−1^. The above measurements were conducted for about twenty-three days. From the gas exchange data, we also calculated the intrinsic water use efficiency (iWUE), which is the A/g_s_ ratio. 

To estimate the change in plant water status the plants experienced during drought, we also measured leaf water potential (Ψ_l_) in non-inoculated and inoculated seedlings under well-watered conditions and after drought-induced stomatal closure. Ψ_l_ was determined in five seedlings per inoculation treatment before and after the drought using a pressure chamber (PMS Instrument Company; Albany, OR, USA). For this purpose, the whole shoot or lateral shoots were wrapped in Saran wrap, excised, and immediately used to determine their Ψ_l_. Due to the small size of the seedlings and the partially destructive nature of the measurement, Ψ_l_ for well-watered seedlings was assessed in different seedlings than those exposed to drought. 

Before initiating the drought experiment, six non-inoculated and six inoculated seedlings were harvested and used to analyze the extent of AMF colonization. Similarly, after completing the drought period, we collected roots from each plant and used them to quantify fungal colonization. Colonization was quantified in all the roots smaller than 2 mm in diameter, which were cut into roughly 2 cm segments. These segments were cleared in 5% KOH for 5 min at 121 °C. Subsequently, the roots were rinsed in water and incubated overnight in a solution containing 0.4 µg mL^−1^ wheat germ agglutinin-horseradish peroxidase (WGA-HRP) and 1% bovine serum albumin in PBS [103]. Samples were then rinsed in PBS and incubated for 3 to 5 min in a VIP HRP substrate (Vector Laboratories, CA, USA). Subsequently, the roots were rinsed in water, mounted on 50% glycerol, and observed through an Olympus BX60 microscope at 200 or 400 magnifications. The observed fungi were grouped into two categories: AMF and septate. For AMF, we quantified the presence of non-septate hyphae (diameter of at least 5 µm), arbuscules, and vesicles. We recorded the presence of hyaline and melanized septate hyphae and microsclerotia for the septate fungi. The different fungal structures and total colonization by the two groups of fungi were quantified by the intersection method [104] with about 150 intersections per sample.

#### 4.1.3. Data Analyses

Differences in colonization between non-inoculated and inoculated seedlings before and after experiencing drought were evaluated using the Wilcoxon rank sum test due to the lack of normality of the data. Summary statistics for colonization are presented as medians and their 95% confidence limits. In plants exposed to drought, the relationship between time since withholding watering and A, Tr, g_s_, and ΦPSII was not linear but followed a negative sigmoidal curve. This response was modeled using a modification of a sigmoidal function used by Guyot et al. [105] to describe the relationship between g_s_ and Ψ_l_. The modified equation is
Response variable = I × (1/(1 + e^(S × (time − t1/2))^))(1)
where the response variable is the particular parameter measured (i.e., A), I is the parameter’s initial value, S is a factor that accounts for the shape of the curve, and t_1/2_ is the time when the parameter reached half its initial value. The data for each plant was fitted to Equation (1) to obtain the values of I, S, and t_1/2_. For this purpose, we used the Non-Linear Least-Square Minimization and Curve-Fitting library (LMFIT) in Python [106]. Appendix A shows an outcome of the curve-fitting process. The values of I and t_1/2_ estimated for each parameter were compared between inoculation treatments by *t*-tests. 

The effect of stomatal conductance and AMF inoculation on iWUE was analyzed using a generalized linear mixed model with a gamma distribution and a log link employing the glmer function in the lmer4 package in R 4.3 [107] (model<-glmer (iWUE~ gs* Inoculation_treatment + (1|plant), data = data, family = Gamma (link = “log”))). We treated conductance and inoculation as fixed factors and individual plants as random factors to account for repeated measurements [108]. In addition, the gamma distribution and log link function allowed us to model the lack of linearity in the g_s_-iWUE relationship and the increase in residual errors with decreases in g_s_ [109]. Comparisons of Ψ_l_ between non-inoculated and inoculated seedlings under well-water conditions and after stomatal closure were made using Welch’s ANOVA. Except for the curve fitting process and the mixed model, all statistical analyzes were implemented using base functions in R 4.3 [110]. *p*-values lower than 0.05 were considered statistically significant in this and the other experiments.

### 4.2. Greenhouse Experiment 2: Inoculation with Soil and Root from the Trap Cultures 

#### 4.2.1. Plant and Fungal Material

The plant material used in this experiment was *A. tridentata* ssp. *wyomingensis* seedlings provided by the Bureau of Land Management. The seedlings were like those this agency uses for outplanting at disturbed sites; they were about ten months old, growing in 150 mL cone-tainers filled with a 3:1 peat moss to vermiculite mix and with a firm root ball [60]. In this experiment, we investigated the possibility of increasing colonization at the transplanting step, mimicking how the seedlings are transplanted in the field but using a larger pot as the transfer site. In addition, we tested a less laborious inoculation method. While inoculation with isolated spores is effective in causing AMF colonization, this procedure is impractical for the many seedlings used in restoration. 

#### 4.2.2. Growing Conditions, Experimental Approach, and Data Collection

Ten months old seedlings were transplanted to 656 mL pots (D40H Deepot Cell, Stuewe & Sons, Inc., OR, USA) filled with a 3:2 sand–soil mix that had been autoclaved twice for 1 h (non-inoculated seedlings) or the same mix blended with potting mix and roots from the trap cultures in a 3 to 1 ratio (inoculated seedlings). The sand–soil mix was identical to that used for producing trap cultures. In addition, we treated pots for the non-inoculated seedlings with an AMF-free microbial wash obtained by blending soil and roots from the trap cultures with water in a 1:15 (*w*/*v*) ratio and passing the suspension through a 35 µm nylon mesh [61]. The filtrate was used to drench the sterilized sand–soil mix using approximately 500 mL per pot and applying half that volume one day and the other half two days later. After transplanting to the pots, plants were grown in a greenhouse for six months under a 15 h photoperiod and day/night conditions of 23/18 ± 3 °C. Plants were fertilized monthly with a 1/8 strength Hoagland’s solution.

Six months after transplanting to the 656 mL pots, the *A. tridentata* seedlings were used to conduct an experiment consisting of two watering treatments (well-watered and drought-stressed) and two inoculation treatments (non-inoculated and inoculated plants). The well-watered plants had four plants per inoculation treatment, and the drought-stressed plants had six. The shoots were about 20 cm tall. The well-watered plants received water to field capacity every other day, while for the drought treatment, drought was imposed by stopping watering. In both the well-watered and drought treatments, a 7.62 cm collar foam covered the top of the pots to minimize evaporative water loss from the soil. In the drought treatment, water was withheld until stomatal conductance was minimal for at least one week. 

We measured A, Tr, g_s_, and ΦPSII every other day, as described under experiment 1, except that the light intensity was 1500 µmol m^−2^ s^−1^. For these plants, preliminary light curves indicated that CO_2_ assimilation was saturated at this light intensity. In addition to the gas exchange and chlorophyll fluorescence measurements, we weighed the pots every other day. Particularly for the drought treatment, we used these values to determine the ratio of the pot weight to the weight at pot capacity (rPC). The latter was the pot weight after watering it to saturation and letting it drain for one day [111]. To correlate changes in rPC with plant water status, we also measured midday Ψ_l_ in randomly selected plants throughout the experiment. 

Colonization was analyzed as described earlier. In addition, at the end of the experiment, we isolated DNA from root fragments of six inoculated plants to ascertain the AMF taxa present in them. Procedures for DNA isolation and amplification of fungal DNA were like those previously described by Serpe et al. [101], except for the primers used for DNA amplification. In the present study, amplicons were produced by nested PCR using the primers LF402F4 (GTGAAATTGTTGAAAGGGAA) and LSUmAr3 (TGCTCTTACTCAAATCTATCAAA) [112] in the first amplification, and the FLR3 (TTGAAAGGGAAACGATTGAAGT) and LR3 (CCGTGTTTCAAGACGGG) in the second one; the latter had been modified by the addition of Illumina adapters at their 5′ ends. The PCR procedure amplifies the D2 region of the 28S ribosomal RNA gene (LSU-D2), producing amplicons for most families in the Glomeromycota [112]. ThePCR products were cleaned using ExoSAP-IT (Applied Biosystems, Waltham, MA, USA), adjusted to a DNA concentration of 20 ng µL^−1^, and sent for sequencing to a commercial facility (Genewiz, Inc., South Plainfield, NJ, USA), where samples were analyzed using an Illumina MiSeq platform that produced paired-end reads (2 × 250 bp). 

#### 4.2.3. Data Analyses

The effect of inoculation on fungal colonization was analyzed as described in experiment 1. To analyze physiological parameters in well-watered plants, we used linear regression to assess whether the parameter values changed during the experiment and differed between the two inoculation treatments. For this purpose, the intercepts and slopes estimated for each parameter were compared between inoculation treatments by *t*-test. The plants’ physiological responses to drought were analyzed using two approaches. One approach was the same as in Experiment 1. We used Equation (1) to estimate the initial value (I) and t_1/2_ for each parameter. These values were then compared between inoculation treatments by *t*-test. The second approach involved analyzing the change in A, Tr, g_s_, and ΦPSII in relation to the ratio of the pot weight to the weight at pot capacity (rPC). The equation used to model this relationship was similar to Equation (1).
Response variable = I × (1/(1 + e^(S × ((1 − rPC) − (1 − rPC1/2))^))(2)
where I is the parameter’s initial value at pot capacity, S is a factor that accounts for the shape of the curve, rPC is the ratio of the pot weight to the weight at pot capacity, and rPC_1/2_ is the value of rPC when the parameter reaches half its initial value. We substrated rPC and rPC_1/2_ from 1 in the equation to model a decrease in the response variable as the abscissa value increases. This approach allowed us to obtain a good data fit using an equation similar to Equation (1) (e.g., Appendix A). The values of rPC_1/2_ estimated after curve fitting were then compared between non-inoculated and inoculated plants by *t*-tests. The effects of g_s_ and inoculation on iWUE were analyzed as described earlier.

The DNA sequences obtained were resolved into amplicon sequence variants (ASV) following the DADA2 pipeline using default settings [113]. The taxonomic identity of the ASVs was inferred to the family or genus level by comparing their sequences to those from the RDP LSU taxonomic training data [62] plus sequences from the phylogenetic reference data for molecular systematic of arbuscular mycorrhizal fungi presented by Krüger et al. [63]. The comparison was made using the assignTaxonomy function in DADA2.

### 4.3. Field Experiment

#### 4.3.1. Fungal and Plant Material, Experimental Approach, and Data Collection

The mycorrhizal inoculum used in this experiment consisted of the soil/sand mix and roots from trap cultures, generated as described in experiment 1. The plant material was *A. tridentata* seedlings with the characteristics described in experiment 2. The study was conducted in Kuna Butte, ID, USA (43°26.161′ N, 116°25.848′ W, 908 m a.s.l.), starting in October 2019. At this time, most of the vegetation at the site was dry and comprised stalks of non-native plants, mainly crested wheatgrass (*Agropyron cristatum*), cheatgrass (*Bromus tectorum*), and tumble mustard (*Sisymbrium altissimum*). Tumble mustard was particularly abundant, which partly determined the site’s selection. We reasoned that the abundance of this non-mycorrhizal plant might reduce the density of AMF present in the soil, increasing the need for inoculation. On 26 October 2019, 300 seedlings were randomly assigned to one of two inoculation treatments, control and inoculated. In the control treatment, the seedlings were planted without inoculum. In contrast, 500 mL of soil and roots from the trap cultures were placed beneath and around each seedling for the inoculated treatment. The seedlings were within a 25 m × 30 m plot forming a grid at a distance of 1 to 1.5 m from each other. All seedlings were watered immediately after outplanting through a PVC tube inserted vertically to about 20 cm from the soil surface. Each seedling was enclosed within a metal tree protector (6 mm mesh and close at the top) [88]. The watering was repeated two weeks later; after that, plants only received natural precipitation. 

We evaluated the effects of inoculation and summer drought on the following response variables: fungal colonization of roots, plant survival, Ψ_l_, g_s_, and percent of plants with flowers. Seedlings for colonization analysis were collected in June and October 2020, eight and twelve months after transplanting. Survival was recorded for two years after outplanting but at different intervals; biweekly or monthly from November 2019 to September 2020 and less frequently afterward. Measurements of midday Ψ_l_ started in mid-summer 2020 and continued to the late summer of 2021. We measured midday Ψ_l_ bi-weekly during the summer and less frequently in fall and spring. Stomatal conductance was measured during the summer of 2021 in the same plants used to measure midday Ψ_l_. Three measurements were made per plant between noon and 2 pm using an SC1 leaf porometer (Meter Group, Pullman, WA, USA). Plants bearing inflorescence were counted in the late summer of 2020 and 2021. A weather station at the site recorded temperature, precipitation, and soil moisture in the top 20 cm of soil (EC-20, ECH_2_O Soil Moisture, Sensor, Meter Group).

#### 4.3.2. Data Analyses

The effect of the inoculation treatment and sample collection time on fungal colonization was analyzed by two-way ANOVA rather than a non-parametric test because the data satisfied the criteria of homoscedasticity and normality of residuals based on the Bartlett and Shapiro–Wilk tests, respectively. For a particular day of measurement, the impact of inoculation on Ψ_l_ and stomatal conductance was estimated using *t*-tests. Possible differences in survival were analyzed using the ggsurvplot function in the Survminer R package [114]. We used a chi-square test to examine the impact of the treatments on the number of plants bearing inflorescence. All statistical analyses were conducted in R 4.3 [110].

## 5. Conclusions

This study indicates that AMF colonization delayed the drought-induced decline in g_s_, CO_2_ assimilation, and ΦPSII, or this decline occurred at a lower soil water content. These effects appeared to be mediated by an AMF increase in soil or plant hydraulic conductivity without affecting the Ψ threshold for stomatal closure. Also, as g_s_ decreased with water stress, iWUE increased more in mycorrhizal plants. Additional carbohydrates generated by prolonging photosynthesis and increasing iWUE could be important in coping with water stress, particularly if they were used to grow fine roots and hyphae toward moist and typically deep soil patches that remain in the soil as summer drought progresses [115]. While the results of the greenhouse experiments revealed responses to AMF colonization that can enhance drought tolerance, their significance in increasing the survival of *A. tridentata* seedlings in nature remains to be tested under more severe drought than the plants experienced in our field experiment. In this regard, possible experiments include planting smaller seedlings that are typically more susceptible to drought [36,41], using rainout shelters to increase water deficits [116], and conducting outplanting over several years or sites to assess AMF effects over a broader range of drought intensities.

## Figures and Tables

**Figure 1 plants-12-02990-f001:**
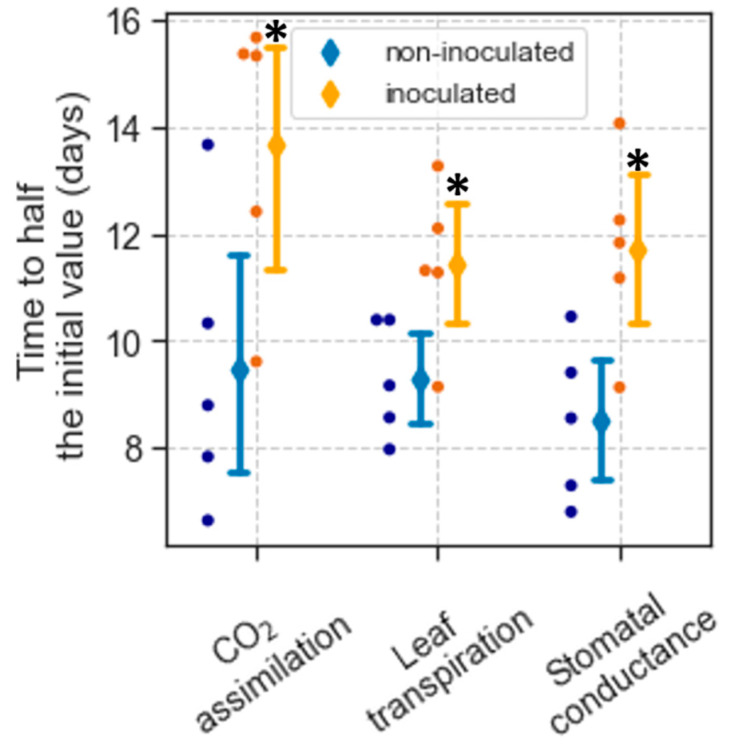
Time for CO_2_ assimilation, leaf transpiration, and stomatal conductance to reach half their initial values after withholding water. Dots show the t_1/2_ of individual seedlings, while diamonds and error bars represent means and 95% confidence limits. For a particular parameter, asterisks indicate statistical differences (*p* < 0.05) between non-inoculated and inoculated seedlings based on *t*-tests.

**Figure 2 plants-12-02990-f002:**
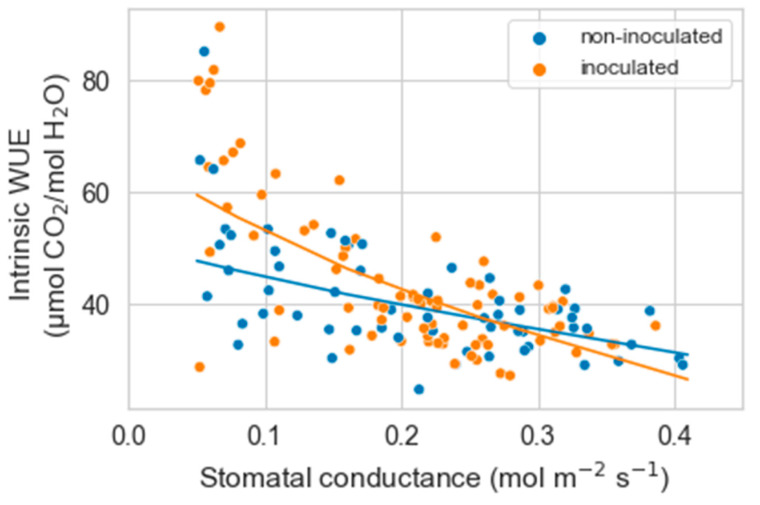
Relationship between stomatal conductance and intrinsic water use efficiency (iWUE) in non-inoculated and inoculated *Artemisia tridentata* seedlings. Based on a generalized linear mixed model, stomatal conductance, inoculation, and their interaction significantly affected iWUE.

**Figure 3 plants-12-02990-f003:**
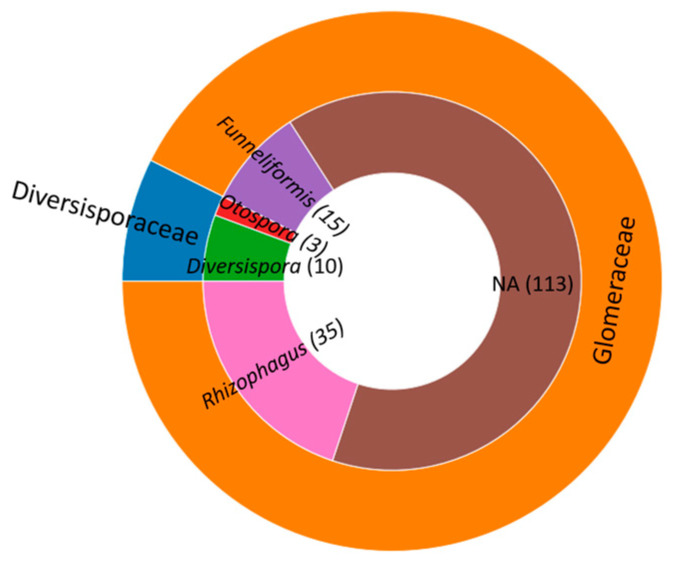
Families and genera of arbuscular mycorrhizal fungi identified in DNA isolated from inoculated *Artemisia tridentata* seedlings. The outer circle indicates families, and the inner circle genera within each family. Numbers in parenthesis specify the amplicon sequence variants detected within each genus or in sequences only identified to the family level (NA). The latter were all within the Glomeraceae.

**Figure 4 plants-12-02990-f004:**
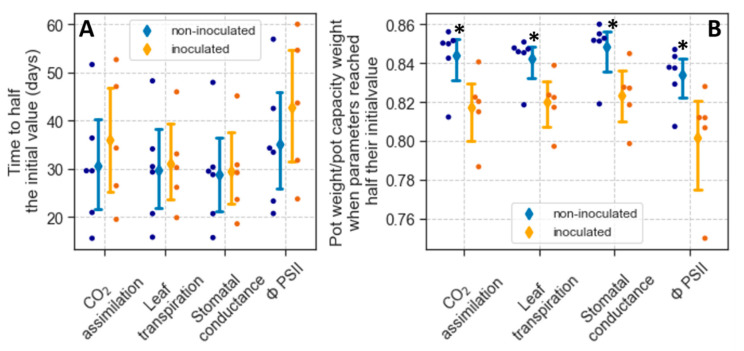
Effect of inoculation on physiological parameters in plants exposed to drought. (**A**) Time for gas exchange and chlorophyll fluorescence parameters to reach half their initial values after withholding water (t_1/2_). (**B**) Ratio of pot weight to weight at pot capacity at which CO_2_ assimilation, leaf transpiration, stomatal conductance, and PSII operating efficiency declined to half their values under well-watered conditions (rPC_1/2_). Dots show the t_1/2_ or rPC_1/2_ of individual seedlings; diamonds and error bars represent means and 95% confidence limits. For a particular parameter, asterisks indicate statistical differences (*p* < 0.05) between non-inoculated and inoculated plants based on *t*-tests.

**Figure 5 plants-12-02990-f005:**
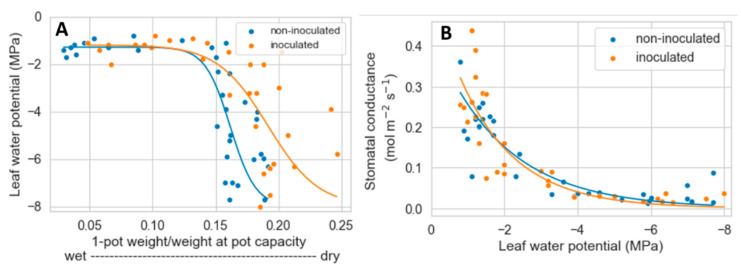
Relationship between decreases in soil water content and plant water potential (**A**) and between leaf water potential and stomatal conductance (**B**) of potted *A. tridentata* plants. Water potential and stomatal conductance measurements were conducted in randomly selected plants throughout the experiment.

**Figure 6 plants-12-02990-f006:**
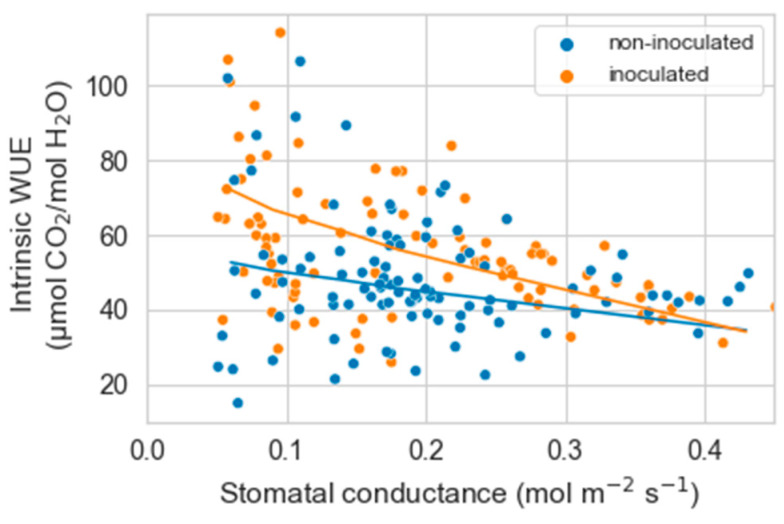
Relationship between stomatal conductance and intrinsic water use efficiency (iWUE) in non-inoculated and inoculated *Artemisia tridentata* seedlings exposed to drought. Based on a generalized linear mixed model, stomatal conductance, inoculation, and their interaction significantly affected iWUE.

**Figure 7 plants-12-02990-f007:**
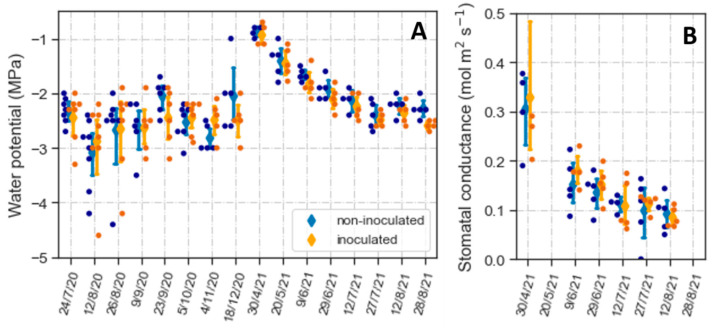
Midday water potential (**A**) and stomatal conductance (**B**) of *Artemisia tridentata* plants outplanted in October 2019. Dots indicate values of individual plants; diamonds and error bars represent means and 95% confidence limits. Differences between non-inoculated and inoculated plants were not significant. Note: To make the data more noticeable, the days when water potential and stomatal conductance were measured are plotted as categorical variables rather than a continuous time sequence.

**Table 1 plants-12-02990-t001:** Percent colonization of *Artemisia tridentata* roots by arbuscular mycorrhizal fungi before and after the imposition of drought. Inoculated plants were grown in a potting mix supplemented with arbuscular mycorrhizal spores. Medians (and 95% confidence limits) of five or six plants. Plants sampled before the drought were different from those after the drought. The *p*-values are based on unpaired Wilcoxon tests.

Colonization	Non-Inoculated	Inoculated	*p*-Value
Total before	0 (0, 18.2)	65.0 (35.6, 75.7)	0.004
Arbuscular before	0 (0, 15.3)	33.8 (14.4, 64.0)	0.015
Vesicles before	0 (0, 0)	9.0 (1.7, 22.2)	0.002
Total after	0 (0, 12.6)	48.2 (39.7, 55.3)	0.008
Arbuscular after	0 (0, 11.5)	16.7 (7.1, 33.8)	0.016
Vesicles after	0 (0, 0)	6.0 (1.2, 21.1)	0.008

**Table 2 plants-12-02990-t002:** Comparison of initial values of CO_2_ assimilation (A), leaf transpiration (Tr), stomatal conductance (g_s_), and operating efficiency of photosystem II (ΦPSII) in non-inoculated and inoculated *Artemisia tridentanta* seedlings before the onset of drought. Mean (±SE) of five plants. Initial values were estimated using Equation (1); *p*-values based on *t*-tests.

Parameter	Non-Inoculated	Inoculated	*p*-Value
A (µmol CO_2_ m^−2^ s^−1^)	10.20 (±1.28)	9.57 (±0.72)	0.68
Tr (mol H_2_O m^−2^ s^−1^)	4.74 (±0.49)	5.61 (±0.42)	0.22
g_s_ (mol H_2_O m^−2^ s^−1^)	0.29 (±0.03)	0.28 (±0.03)	0.77
ΦPSII	0.46 (±0.014)	0.45 (±0.015)	0.54

**Table 3 plants-12-02990-t003:** Percent colonization of *Artemisia tridentata* roots by arbuscular mycorrhizal fungi before and after the imposition of drought and in plants always kept well-watered. Medians (and 95% confidence limits) of four to six plants. Plants sampled before the drought were different from those after the drought. The *p*-values are based on unpaired Wilcoxon tests.

Colonization	Non-Inoculated	Inoculated	*p*-Value
Total before	0 (0, 0.9)	20.6 (14.3, 22.3)	0.008
Arbuscular before	0 (0, 0.8)	6.9 (1.8, 14.3)	0.007
Vesicles before	0 (0, 0)	2.0 (0.9, 2.9)	0.005
Total after	1.5 (0, 4.9)	32.3 (17.0, 43.7	0.004
Arbuscular after	0 (0, 1.8)	6.3 (0, 14.6)	0.03
Vesicles after	0 (0, 0.4)	2.0 (1.0, 2.1)	0.004
Total well-watered	1.0 (0, 4.5)	31.3 (22.1, 42.6)	0.029
Arbuscular well-watered	0 (0, 0.9)	17.5 (13.5, 30.3)	0.029
Vesicles well-watered	0 (0, 0)	2.1 (0.5, 4.6)	0.014

**Table 4 plants-12-02990-t004:** Comparison of initial values of CO_2_ assimilation (A), leaf transpiration (Tr), stomatal conductance (g_s_), and operating efficiency of photosystem II (ΦPSII) in non-inoculated and inoculated *Artemisia tridentanta* plants before the onset of drought. Mean (±SE) of five or six plants. Initial values were estimated using Equation (1); *p*-values based on *t*-tests.

Parameter	Non-Inoculated	Inoculated	*p*-Value
A (µmol CO_2_ m^−2^ s^−1^)	11.4 (±1.8)	11.9 (±1.9)	0.84
Tr (mol H_2_O m^−2^ s^−1^)	5.3 (±0.6)	4.9 (±0.4)	0.62
g_s_ (mol H_2_O m^−2^ s^−1^)	0.27 (±0.04)	0.26 (±0.03)	0.87
ΦPSII	0.21 (±0.01)	0.20 (±0.01)	0.42

**Table 5 plants-12-02990-t005:** Percent colonization of field-grown *Artemisia tridentata* plants by arbuscular mycorrhizal fungi. Inoculated plants were supplemented with soil and roots from trap cultures at transplanting. Plants were harvested eight (spring) or twelve (fall) months after outplanting. Average and standard errors of eight (spring) or four (fall) plants.

Colonization	Non-Inoculated	Inoculated	*p*-Value
Total spring	24.9 (±4.8)	46.7 (±4.8)	0.008
Arbuscular spring	7.2 (±2.9)	17.0 (±2.9)	0.027
Vesicles spring	8.1 (±3.1)	8.4 (±3.1)	0.950
Total fall	8.4 (±6.7)	33.8 (±6.7)	0.028
Arbuscular fall	3.0 (±4.1)	9.2 (±4.1)	0.302
Vesicles fall	0.3 (±5.5)	11.8 (±5.5)	0.1892

## Data Availability

The data presented in this study are available within the article and its Appendix A.

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
