# Peer review of "Arbuscular Mycorrhizae Alter Photosynthetic Responses to Drought in Seedlings of Artemisia tridentata"

_plants, 2023, doi:10.3390/plants12162990_

Round 1

Reviewer 1 Report

The manuscript by Geisler and collaborators aims at investigating the possible effect of symbioses with arbuscular mycorrhizal fungi (AMF) to help plants of Artemisia tridentata, a keystone species of the sagebrush steppe, to cope with drought. To fill this aim, the authors set up 3 experiments, 2 in greenhouse and 1 in field conditions, each experiment differing regarding the mode of AMF inoculation with native AMF. In the first experiment, 2-month-old seedlings were inoculated with an aqueous suspension containing about 700 spores extracted from trap cultures to get mycorrhizal plants. In the second experiment, 10-month-old seedlings were inoculated using the potting substrate and roots from trap cultures instead of spores only. In the third experiment, at the time of plantation in the field, 500 ml of soil and roots from the trap cultures were placed beneath and around each 10-month-old seedling for the inoculated treatment. Seedlings were then exposed to drought, either in greenhouse or in the field. AMF colonization was higher in roots of plants grown in greenhouse than in field conditions. Also, the drought stress was not controlled in the field experiment and was not strong enough to reveal any effect of AMF on the survival rates of plants. However, in greenhouse conditions, AMF colonization increased intrinsic water use efficiency under water stress and delayed the  decrease in photosynthesis caused by drought, this decrease occurring at a lower soil water content.

The manuscript is very well written and I am sure that the data it contains will be of great interest for the people working on AMF symbiosis. I have only minor comments that are given below.

Comments

- In the Introduction, lines 26-27, I would have cited the paper from Brundrett & Tedersoo  (2018) (Evolutionary history of mycorrhizal symbioses and global host plant diversity. New Phytologist 220, 1108–1115. doi:10.1111/nph.14976) which is more recent than Wang & Qiu (2006). The Brundrett & Tedersoo’s paper indicates that true AM symbiosis is formed with 72% of plant species, not 80%. If the authors agree with my comment, could they modify their sentence as well as the following ones?

- Line 65, add “with” between “but generally” and “low success rates”

- Lines 78-79, the authors wrote “To better understand the effect of AMF on A. tridentata seedlings' responses to drought, we conducted two greenhouse experiments”. But I think this sentence is misleading for the reader because 3 experiments were carried out, not 2. I would have written “To understand better the effect of AMF on A. tridentata seedlings' responses to drought, we conducted three experiments, two in greenhouse and one in field conditions”.

- In the Results section, the authors start each subpart  (3.1; 3.2; 3.3) by giving a summary of the methods used (Lines 103-112; 206-218; 341-349) but I think it is not necessary as it is  Materials and Methods.

- Line 728, the authors wrote “The filtrate was used to drench the sterilized  sand-soil mix” but it would be useful to know the volume of filtrate that was used for each plant.

Author Response

Below, we included the reviewer's comments and the changes made to the manuscript in response to the comments. The changes made, and other answers to the reviewer's comments are in bold font and highlighted in yellow. We thank the reviewer for the valuable input.

Reviewer 1

The manuscript by Geisler and collaborators aims at investigating the possible effect of symbioses with arbuscular mycorrhizal fungi (AMF) to help plants of Artemisia tridentata, a keystone species of the sagebrush steppe, to cope with drought. To fill this aim, the authors set up 3 experiments, 2 in greenhouse and 1 in field conditions, each experiment differing regarding the mode of AMF inoculation with native AMF. In the first experiment, 2-month-old seedlings were inoculated with an aqueous suspension containing about 700 spores extracted from trap cultures to get mycorrhizal plants. In the second experiment, 10-month-old seedlings were inoculated using the potting substrate and roots from trap cultures instead of spores only. In the third experiment, at the time of plantation in the field, 500 ml of soil and roots from the trap cultures were placed beneath and around each 10-month-old seedling for the inoculated treatment. Seedlings were then exposed to drought, either in greenhouse or in the field. AMF colonization was higher in roots of plants grown in greenhouse than in field conditions. Also, the drought stress was not controlled in the field experiment and was not strong enough to reveal any effect of AMF on the survival rates of plants. However, in greenhouse conditions, AMF colonization increased intrinsic water use efficiency under water stress and delayed the  decrease in photosynthesis caused by drought, this decrease occurring at a lower soil water content.

The manuscript is very well written and I am sure that the data it contains will be of great interest for the people working on AMF symbiosis. I have only minor comments that are given below.

Comments

- In the Introduction, lines 26-27, I would have cited the paper from Brundrett & Tedersoo  (2018) (Evolutionary history of mycorrhizal symbioses and global host plant diversity. New Phytologist 220, 1108–1115. doi:10.1111/nph.14976) which is more recent than Wang & Qiu (2006). The Brundrett & Tedersoo's paper indicates that true AM symbiosis is formed with 72% of plant species, not 80%. If the

We replaced the reference and the estimate of plant species that form associations with AMF as indicated by the reviewer (line 27 in the revised manuscript). Thank you for reminding us of the Brundett & Tedesoroo paper!

  • Line 65, add "with" between "but generally" and "low success rates"

We changed the sentence to make it more clear (line 73-74).

- Lines 78-79, the authors wrote "To better understand the effect of AMF on A. tridentata seedlings' responses to drought, we conducted two greenhouse experiments". But I think this sentence is misleading for the reader because 3 experiments were carried out, not 2. I would have written "To understand better the effect of AMF on A. tridentata seedlings' responses to drought, we conducted three experiments, two in greenhouse and one in field conditions".

We modified the sentence as suggested by the reviewer; the reviewer's version is more accurate. (line 88-89). We also added information to the paragraph based on comments from the second reviewer.

 - In the Results section, the authors start each subpart  (3.1; 3.2; 3.3) by giving a summary of the methods used (Lines 103-112; 206-218; 341-349) but I think it is not necessary as it is  Materials and Methods.

In the Results, we summarized the methods for each experiment due to the format followed by Plants, where Results precede Materials and Methods. We agree with the reviewer that the indicated subparts are not entirely necessary. If the reviewer or editor prefers, we can delete the indicated paragraphs. However, we think leaving them makes the manuscript clearer or easier to read. To condense the paragraphs further, we moved a sentence for the first subpart to the Materials and Methods (lines 669-670), shortened a couple of sentences in the second subpart, and deleted one sentence from the third subpart.

- Line 728, the authors wrote "The filtrate was used to drench the sterilized  sand-soil mix" but it would be useful to know the volume of filtrate that was used for each plant.

Done, we indicated the volume of filtrate poured into each pot. (lines 788 -790)

Reviewer 2 Report

I read the manuscript Plants- 2562191 with great  interest.

The ms. report on the investigation of plant-physiological changes of Artemisia tridentata with arbuscular mycorrhizae exposed to drought in greenhouse and field.

overall, I found the manuscript to be well-written and understandable, I have only some minor questions which intrigued me

-        Introduction:

line 93-100: I suggest to explain better the informations of fungal endophytes that may be present in A. tridentata root and may alter the effect of AMF.

-        Results:

line 248-256: I suggest to summarize the results of the amplicon sequencing at genus-level in a diagram.

-        Discussion:

Does the genus composition of AMF affect the examined physiological results?

How can stimulate AMF the phytohormone control? Could this be a reason for the increased number of inflorescences in the case of AMF colonized plants? Please explain the phytohormone background of AMF colonization, if it has significance.

-        Materials and methods:

Can you explain in detail the Illumina sequencing methodology?

-        Conclusion:

Please describe what further studies are planned to further support the hypothesis, especially in field conditions.

Author Response

Below, we included the reviewer's comments and the changes made to the manuscript in response to the comments. The changes made, and other answers to the reviewer's comments are in bold font and highlighted in yellow. We thank the reviewer for the valuable input. 

Reviewer 2

I read the manuscript Plants- 2562191 with great  interest.

The ms. report on the investigation of plant-physiological changes of Artemisia tridentata with arbuscular mycorrhizae exposed to drought in greenhouse and field.

Overall, I found the manuscript to be well-written and understandable, I have only some minor questions which intrigued me

-        Introduction:

line 93-100: I suggest to explain better the informations of fungal endophytes that may be present in A. tridentata root and may alter the effect of AMF.

We added various sentences to the last paragraph of the discussion to describe the endophytic septate fungi that might be present in A. tridentata roots and their possible effects on the plants, including specific examples of how the occurrence of septate fungi may affect the results or complicate their interpretation (lines 111-126). Also, a paragraph was added to the discussion to argue that colonization by septate fungi cannot account for the differential responses to drought between non-inoculated and AMF-inoculated plants (lines 476-484).  

-        Results:

line 248-256: I suggest to summarize the results of the amplicon sequencing at genus-level in a diagram.

We made a nested pie chart showing the family, genera, and the number of ASVs within each of them (Figure 3, lines 290-297).

-        Discussion:

Does the genus composition of AMF affect the examined physiological results?

Yes, the genera and even the strain within a specific AMF species could affect the results. Although AMF have low host specificity and are typically mutualistic,  their effects on plants vary depending on the particular AMF-plant taxa involved in the association. In this regard, an important factor is whether the AMF associated with the plant is native or exotic; native AMF tend to be more beneficial than exotic ones. For this reason, we multiplied and used native AMF in our study. We added a few sentences to the introduction to present these concepts and better justify using native AMF (lines 90-96). Thank you for bringing up this topic!

Our study was not designed to analyze the effect of different AMF taxa on the measured responses. Also,  to our knowledge, one cannot generalize that particular AMF genera are more beneficial than others. Consequently,  discussing the results in relation to the different genera present seems, at this stage, too speculative. It certainly would be interesting to evaluate the effect of different taxa on photosynthesis and other physiological parameters; we are currently trying to produce cultures of single species of native AMF for this purpose.

How can stimulate AMF the phytohormone control? Could this be a reason for the increased number of inflorescences in the case of AMF colonized plants? Please explain the phytohormone background of AMF colonization, if it has significance.

An effect of AMF on the hormonal balance of the plants could be a reason for the increase in the number of plants that developed inflorescences. However, we searched the literature and found no references linking AMF-induced hormone changes to the transition to flowering; most articles attributed effects on flowering to plant nutrition. Several studies and reviews indicate that AMF can improve reproductive fitness by affecting hormone balance, but this effect is mainly by enhancing pollen production and viability rather than initiating flowering. Nevertheless, we added a couple of sentences and literature to the discussion to mention that hormones could have mediated the effect on inflorescence development (lines 637-642).

We are familiar with the molecular details and hormonal interactions responsible for initiating AMF colonization, and we think it is a fascinating scientific story. However, we do not have data to relate the crosstalk of signals during colonization to the observed flowering phenomena. Also, we did not find any reference in the literature connecting these processes. Thus, we prefer not to discuss these events further.

-        Materials and methods:

Can you explain in detail the Illumina sequencing methodology?

We added information to the M&M to explain the analysis was conducted using an Illumina MiSeq platform that produced paired-end reads (lines 826-827).

-        Conclusion:

Please describe what further studies are planned to further support the hypothesis, especially in field conditions.

A sentence was added to the conclusions to describe probable future experiments (lines 917-920).

Reviewer 3 Report

The authors have compiled an interesting piece of work entitled ‘Arbuscular mycorrhizae alter photosynthetic responses to drought in seedlings of Artemisia tridentata’. Although author have briefly covered the different aspect.

However author should check and resolve the minor quarries

Line- 36 Please details  some mechanism and responses  ‘For drought, in particular, results indicate that AMF can increase plant drought tolerance through various mechanisms and responses…..

Line 50- that overall, AMF increases gs,?? Write some details about gs

 Line 602 - The spores were suspended in water and used as inoculum  within a month. Only water?

Line 99- What author mean for ……we quantified their occurrence as an additional factor that  could affect the results.

Check sentence 113- We measured colonization before and after the imposition of drought but in different  plants due to the destructive nature of the measurements

Line 107-108 authors have   mentioned  inoculation of  700 spores. Did they  have used  any markers, that confirm inoculation 

The authors have compiled an interesting piece of work entitled ‘Arbuscular mycorrhizae alter photosynthetic responses to drought in seedlings of Artemisia tridentata’. Although author have briefly covered the different aspect.

However author should check and resolve the minor quarries

Line- 36 Please details  some mechanism and responses  ‘For drought, in particular, results indicate that AMF can increase plant drought tolerance through various mechanisms and responses…..

Line 50- that overall, AMF increases gs,?? Write some details about gs

 Line 602 - The spores were suspended in water and used as inoculum  within a month. Only water?

Line 99- What author mean for ……we quantified their occurrence as an additional factor that  could affect the results.

Check sentence 113- We measured colonization before and after the imposition of drought but in different  plants due to the destructive nature of the measurements

Line 107-108 authors have   mentioned  inoculation of  700 spores. Did they  have used  any markers, that confirm inoculation 

Author Response

Below, we included the reviewer's comments and the changes made to the manuscript in response to the comments. The changes made, and other answers to the reviewer's comments are in bold font and highlighted in yellow. We thank the reviewer for the valuable input.

Comments and Suggestions for Authors

The authors have compiled an interesting piece of work entitled 'Arbuscular mycorrhizae alter photosynthetic responses to drought in seedlings of Artemisia tridentata'. Although author have briefly covered the different aspect.

However author should check and resolve the minor quarries

Line- 36 Please details  some mechanism and responses' For drought, in particular, results indicate that AMF can increase plant drought tolerance through various mechanisms and responses…..

Some mechanisms and responses were described in the same paragraph. We have modified the paragraph to clarify this (lines 36-53).

Line 50- that overall, AMF increases gs,?? Write some details about gs

We added more details regarding the effect of AMF on gs (lines 55-59).

 Line 602 - The spores were suspended in water and used as inoculum  within a month. Only water?

Sorry, we meant to say that the spores remained in the antibiotic solution until the day of inoculation, and that on that day, we resuspended the spores in water. The sentence was corrected (lines 659-660). Thanks for noticing this!

Line 99- What author mean for ……we quantified their occurrence as an additional factor that  could affect the results.

We expanded the last paragraph of the introduction to give examples of how the presence of septate fungi may affect the results (lines 111-126).

Check sentence 113- We measured colonization before and after the imposition of drought but in different  plants due to the destructive nature of the measurements

A sentence was added to explain in more detail what we meant by destructive measurements (lines 137-140).

Line 107-108 authors have   mentioned  inoculation of  700 spores. Did they  have used  any markers, that confirm inoculation 

We are not sure what the reviewer means by markers to check inoculation. We checked that inoculation was effective by measuring AMF colonization before and after the imposition of drought. The high levels of AMF colonization in inoculated seedlings compared with the non-inoculated ones indicate that the added spores germinated and colonized the roots.